# Validity and Absolute Reliability of the Cobb Angle in Idiopathic Scoliosis with TraumaMeter Software

**DOI:** 10.3390/ijerph19084655

**Published:** 2022-04-12

**Authors:** José Hurtado-Avilés, Fernando Santonja-Medina, Vicente J. León-Muñoz, Pilar Sainz de Baranda, Mónica Collazo-Diéguez, Mercedes Cabañero-Castillo, Ana B. Ponce-Garrido, Victoria Eugenia Fuentes-Santos, Fernando Santonja-Renedo, Miriam González-Ballester, Francisco Javier Sánchez-Martínez, Pietro Gino Fiorita, Jose Manuel Sanz-Mengibar, Joaquín Alcaraz-Belzunces, Vicente Ferrer-López, Pilar Andújar-Ortuño

**Affiliations:** 1Sports & Musculoskeletal System Research Group (RAQUIS), University of Murcia, 30100 Murcia, Spain; joseaviles@um.es (J.H.-A.); fernando@santonjatrauma.es (F.S.-M.); psainzdebaranda@um.es (P.S.d.B.); monicacodi@hotmail.com (M.C.-D.); vefs1204@gmail.com (V.E.F.-S.); jmsmengibar@hotmail.com (J.M.S.-M.); pilarandujar.albacete@gmail.com (P.A.-O.); 2Department of Orthopaedic Surgery and Traumatology, Hospital Clínico Universitario Virgen de la Arrixaca, 30120 Murcia, Spain; 3Department of Surgery, Pediatrics and Obstetrics & Gynecology, Faculty of Medicine, University of Murcia, 30100 Murcia, Spain; 4Department of Orthopaedic Surgery and Traumatology, Hospital General Universitario Reina Sofía, 30003 Murcia, Spain; 5Department of Physical Activity and Sport, Faculty of Sport Sciences, Regional Campus of International Excellence “Campus Mare Nostrum”, University of Murcia, 30100 Murcia, Spain; 6Department of Rehabilitation Sciences and Physiotherapy, Albacete University Hospital Complex, 02006 Albacete, Spain; 7Department of Rehabilitation Sciences and Physiotherapy, General Hospital, Almansa, 02640 Albacete, Spain; albacetemercedes@gmail.com (M.C.-C.); anabelenponga@yahoo.es (A.B.P.-G.); 8“La Vega Lorenzo Guirao” Hospital, 30530 Cieza, Spain; fsr88@hotmail.com; 9Medicine Faculty, University of Murcia, 30100 Murcia, Spain; mgfisiofuncional@gmail.com (M.G.-B.); franj.sanchez@hotmail.com (F.J.S.-M.); pietro.f.95@live.it (P.G.F.); joalbe81@hotmail.com (J.A.-B.); 10Centre for Neuromuscular Diseases, National Hospital for Neurology and Neurosurgery, University College London Hospitals NHS Foundation Trust, London WC1N 3BG, UK; 11Department of Physiotherapy, Faculty of Medicine, University of Murcia, 30100 Murcia, Spain; ferrerlopezv@gmail.com

**Keywords:** spine, adolescent idiopathic scoliosis, Cobb angle, measurement, software applications, validity, reliability

## Abstract

The Cobb angle value is a critical parameter for evaluating adolescent idiopathic scoliosis (AIS) patients. This study aimed to evaluate a software’s validity and absolute reliability to determine the Cobb angle in AIS digital X-rays, with two different degrees of experienced observers. Four experts and four novice evaluators measured 35 scoliotic curves with the software on three separate occasions, one month apart. The observers re-measured the same radiographic studies on three separate occasions three months later but on conventional X-ray films. The differences between the mean bias errors (MBE) within the experience groups were statistically significant between the experts (software) and novices (manual) (*p* < 0.001) and between the novices (software) and novices (manual) (*p* = 0.005). When measured with the software, the intra-group error in the expert group was MBE = 1.71 ± 0.61° and the intraclass correlation coefficient (ICC (2,1)) = 0.986, and in the novice group, MBE = 1.9 ± 0.67° and ICC (2,1) = 0.97. There was almost a perfect concordance among the two measurement methods, ICC (2,1) = 0.998 and minimum detectable change (MCD95) < 0.4°. Control of the intrinsic error sources enabled obtaining inter- and intra-observer MDC95 < 0.5° in the two experience groups and with the two measurement methods. The computer-aided software TraumaMeter increases the validity and reliability of Cobb angle measurements concerning manual measurement.

## 1. Introduction

Adolescent idiopathic scoliosis (AIS) is a three-dimensional deformity involving the axial, sagittal, and frontal planes [1]. AIS can progress over the years, especially during growth, and can cause musculoskeletal, lung, and psychological problems and significant pain in adulthood [2]. The Cobb angle (described by John Robert Cobb in 1948) measurement on the standing posteroanterior full-length spine X-ray is the gold standard for diagnosing and monitoring AIS changes [3]. Cobb angle measurement is necessary to assess the severity of scoliosis and to quantify the risk of progression [4,5,6,7], for the selection of treatment [3,5,8], and the analysis of orthopaedic and surgical procedures [3,4,6,9,10,11] and the effectiveness of treatment [12,13,14]. The Cobb angle is the most important measurement next to vertebral rotation on AIS radiographs [3,4], as it is necessary to establish the diagnosis and decide on the treatment. A sufficiently large error in the Cobb angle measurement can mean, for example, that the indicated treatment varies from observation to orthopaedic therapy or from bracing to instrumented arthrodesis.

Using a computerised tomography (CT) scan, a three-dimensional reconstruction of the spine can be obtained to quantify AIS with a high level of accuracy [1]. However, the CT scan is not suitable for monitoring scoliotic progression because of the excessive and repeated radiation (e.g., an estimated radiation dose of 5.2 mSv for each study [15]). Radiographic medical imaging, especially the standing posteroanterior full-length spine X-ray [16,17,18], continues to be the method of choice for diagnosing and monitoring scoliosis [19].

Traditionally, an increase in the Cobb angle of 5° between successive measurements indicates scoliosis progression [12,20,21,22,23]. Although conventional Cobb angle scoliosis measurement is a simple technique, there are numerous studies of manual measurements of Cobb angle with an average inter-observer variability greater than 5° [12,13,24,25,26,27,28,29,30]. Potential sources of intrinsic error for Cobb angle measurement are poor-quality digital X-rays images, the incorrect definition of the cranial and caudal vertebrae, variable width markers/pencils, different protractors, inaccurate drawing of the lines along the vertebral endplates, imprecise drawing of perpendicular lines, and inaccurate angle measurement itself [12,17,30,31,32,33,34,35,36]. Since the establishment of digital medical imaging, several authors have developed computer-assisted measurement systems to measure the Cobb angle on digital AIS images. These computer programmes avoid sources of intrinsic error compared to conventional measurement on X-ray films [17,28,31,34,36,37,38,39,40]. Using these systems, different authors reported intra-observer MBE between 1° and 2° [31,34,37,38] and between 2° and 4° [17,28,36].

The present study aims to: (1) evaluate the intra- and inter-observer absolute reliability and validity of a computer-aided Cobb measurement method designed to reproduce the manual Cobb method in AIS digital images, focused on reducing intrinsic error sources in two groups of observers, novices and experts; (2) investigate if the developed software is sensitive to observer skill levels or experiences; and (3) compare the software method with the manual Cobb method. We have hypothesised that the intra-observer error in Cobb angle measurements is less than 2.5° in novices and even better (less than 2°) in experts and that the use of the software TraumaMeter improves the validity and reliability of the measurements obtained compared to the manual method.

## 2. Materials and Methods

### 2.1. Software

We developed a computer-aided measurement system (TraumaMeter v.873, José Hurtado Avilés and Fernando Santonja Medina, registration number 08/2021/374, Murcia, Spain) that digitally reproduces the manual Cobb angle measurement method on digital X-ray images [41,42]. The software was developed in C++ language under the Microsoft Visual Studio 2019 (version 16.3.5, Microsoft Corporation, Redmond, WA, USA) development environment using the OpenCV 3.4.10 (Intel Corporation, Santa Clara, CA, USA) artificial vision libraries and the DCMTK libraries, from OFFIS, Institute for Information Technology, to operate with DICOM (digital imaging and communication on medicine) files. The software incorporates additional tools, such as the ability to zoom in on regions of interest and to vary the contrast (fractional difference in optical density of the brightness between two regions of an image) of the digitalised X-ray image.

The system allows the evaluator to choose several cranial and caudal vertebrae from the curve, with the software selecting the most tilted ones, returning a Cobb angle result expressed in degrees, as shown in Figure 1. To measure the Cobb angle, the observer opens the X-ray image, enlarges the vertebra, and selects with a mouse click the two points defining the lines tangent to the cranial endplate of the curve and the two caudal endplate points.

### 2.2. Study Design and Measurement Protocol

The validity and reliability of the traditional manual measurement method were studied to validate the software, focusing on decreasing the sources of intrinsic measurement error. We conducted a prospective and observational study of 35 scoliotic curves in 21 selected standing frontal full-length spine X-rays of patients with AIS. The X-ray sample was homogeneous, had equivalent image quality, and had no defects.

The radiographic images were collected from an image repository in a retrospective manner during the routine medical care of patients with AIS. Our study followed the World Medical Association Declaration of Helsinki’s ethical standards, as revised in 2013. The study was granted exemption from requiring ethics approval since the complete and irreversible anonymisation of the images did not involve data processing. The X-ray images were obtained natively in digital format (in DICOM, with a resolution of 283.46 pixels/mm) and printed in 350 × 430 mm format.

The selected X-rays showed, according to the angular classification proposed by the International Society on Scoliosis Orthopaedic and Rehabilitation Treatment [43]: low scoliosis in 9 cases (curves between 11° to 20°), moderate scoliosis in 11 cases (curves between 21° and 35°), moderate to severe scoliosis in 6 cases (curves between 36° and 40°), severe scoliosis in 4 cases (curves between 41° and 50°), severe to very severe scoliosis in 3 cases (curves between 51° and 55°), and very severe scoliosis in 2 cases (curves with 56° or more).

We assessed absolute reliability according to the Hopkins criteria (minimum n of 30 cases, at least six blinded observers as assessors, and at least three tests per observer, separated by at least two weeks) [44,45]. We also assessed validity.

The research was carried out with eight independent evaluators with different experience levels in measuring Cobb angles. Four observers, considered “Experts”, were an orthopaedic specialist and three physical therapy and rehabilitation specialists who are accustomed to measuring spinal misalignments in their daily practice. The division of the observers into the expert and novice groups was made based on the frequency with which they use the Cobb method rather than based on their medical speciality. We considered experts to be observers that were very often involved in the follow-up and monitoring of patients with AIS. Four “novice” observers were professionals from different health sciences branches (not orthopaedists) and who, although they knew the theory of how to make measurements on X-rays of the spine, had never measured with Cobb’s method.

In each of the 21 X-rays, each observer identified the primary curve and the secondary or compensatory curve and measured them with the software on three occasions separated by one month (Appendix Aa). To validate the software, the observers re-measured the same radiographic studies three months later but on X-ray films (analogical radiographs) in a conventional manual way (Appendix Ab). The conventional measurement was also repeated on three occasions, one month apart. To avoid bias, the sequence in which the radiographs were presented was randomly assigned in each measurement round by the study coordinator, who kept the randomisation key confidential. In total, 1680 Cobb angles were measured for this study (210 by each observer).

A 5 h briefing was held before the software TraumaMeter v.873 measurements, with comprehensive information on the study and training in software use. Similarly, one month after completing the measurements with the software and before the manual measurements, a briefing session was held with Cobb’s method’s relevant indications for the correct measurement. The observers received the 21 X-ray films, the same kind of ruler, square, bevel, permanent black fine-point ink marker, and the same protractor and transparent acetate sheets for the manual measurements to mark the reference points and measure without leaving any marks or signals on the X-ray images that could alter the results of the investigation. 

### 2.3. Statistics

Statistical analysis was performed using the Statistical Package for the Social Sciences (SPSS), version 25 for Windows (SPSS, Inc., Chicago, IL, USA). The results were rounded to one decimal place in the measurements obtained with the software and obtained with one decimal place in the manual conventional measurement due to the scale of each measuring instrument. The average of the errors at each retest of the four observers in each group was employed to estimate the agreement between each experience group and the different experience groups.

The distributions of measurements for each curve and the error distributions were improved by identifying values lower than Q1 − 1.5 × IQR (interquartile range)) and higher than Q3 + (1.5 × IQR). These values were considered outliers and were eliminated from each distribution. We removed outliers because of their effect on the normality loss in the data distributions. To be able to apply statistical inference methods, these distributions must be sufficiently normal. Appendix Aa–c show the outliers removed from each distribution.

The Shapiro–Wilk test was used to check that the *p*-values of the data were above the significance level of 0.05, with the null hypothesis that the data fit a normal distribution being accepted. All distributions met the normality criterion of this test.

We used the 24-measurement mean obtained from the three measurements made by each of the eight assessors for each curve and each method to assess both methods’ concordance.

To analyse intra- and inter-group agreement of the software and manual measurements, we calculated the validity (MBE, mean bias error), the reliability (SD, standard deviation), the standard error of the sample (SE), the minimum detectable change (MDC95), and the intra-class correlation coefficient of absolute concordance using a two-factor random-effects model (ICC (2,1)) [46]. We assessed the intra- and inter-observer reliability according to the criteria by Landis and Koch (<0 indicate no agreement, 0.00 to 0.20 indicate slight agreement, 0.21 to 0.40 indicate fair agreement, 0.41 to 0.60 indicate moderate agreement, 0.61 to 0.80 indicate substantial agreement, and 0.81 to 1.0 indicate almost perfect or perfect agreement) [47]. Although the Landis and Koch criteria are for qualitative estimates, we consider that this criterion can serve as a reference for quantitative determinations by measuring the same thing, i.e., the degree of concordance. We also obtained the Bland–Altmann plot for the agreement between the analysis of the manual and software measurement methods.

We analysed whether the differences in MBE values between each set of measurements were statistically significant using ANOVA and Tukey’s method for multiple comparisons. Student’s *t*-test for independent samples was used to analyse the two intergroup distributions (obtained with the software and manually).

## 3. Results

In the ANOVA for the intra-group distributions, we obtain a *p* < 0.001, and, according to Tukey’s method, the differences between the MBE values are statistically significant at a confidence level of at least 95% between the expert (software) and novice (manual) groups (*p* < 0.001) and between the novice (software) and novice (manual) groups (*p* = 0.005) (Figure 2).

As Figure 3 shows, the MBE value of the two inter-group distributions obtained with the software and manually is different when using TraumaMeter or the manual method (*p* < 0.001).

The Table 1 shows the validity and reliability of the intra- and inter-group measures obtained with both measurement methods.

When measuring with the software, the intra-group error in the expert group was MBE = 1.71°, SD = 0.61°, ICC (2,1) = 0.986 (95% CI: 0.977–0.992) and in the novice group was MBE = 1.9°, SD = 0.67°, ICC (2,1) = 0.97 (95% CI: 0.95–0.985). When measured manually, the intra-group error in the expert group was MBE = 2.13°, SD = 0.75°, ICC (2,1) = 0.981 (95% CI: 0.97–0.99) and in the novice group was MBE = 2.50°, SD = 0.88°, ICC (2,1) = 0.974 (95% CI: 0.954–0.988).

The mean intra-observer error with the software was MBE = 1.8°, SD = 0.65° and when measuring manually was MBE = 2.31°, SD = 0.83°.

In the inter-group study (experts versus novices), when measuring manually the error was MBE = 2.47°, SD = 0.76°, ICC (2,1) = 0.973 (95% CI: 0.951–0.988), and when measuring with the software the error was MBE = 1.82°, SD = 0.59°, ICC (2,1) = 0.973 (95% CI: 0.954–0.987).

The evaluation of the agreement between both measurement methods showed that MBE = 0.08°, SD = 0.844°, SEM = 0.143°, MCD95 = 0.395° and an ICC (2,1) = 0.998 (95% CI: 0.996–0.999). The Bland–Altman graphical representation shows the absence of bias in both method agreements (Figure 4).

## 4. Discussion

Our research demonstrates that TraumaMeter software allows an inexperienced observer to measure the Cobb angle with high validity and reliability. We have developed and evaluated a computer-aided measurement system that allows a reduction in the factors responsible for the intrinsic error of the traditional manual measurement, such as the selection of the reference points on the vertebral bodies, inaccurate drawing of the lines along the vertebral endplates, the determination of the perpendicular, and the measurement of the angle. Another robustness of the present study is the use groups of observers with different grades of experience. According to our results, no significant improvement is attributable to practice. Neither with the software (ICC equal to 0.979 in the first evaluation, 0.977 in the second, and 0.982 in the third) nor with the manual method (ICC equal to 0.975 in the first evaluation, 0.977 in the second, and 0.978 in the third).

There is a consensus in the literature that the difference between Cobb angle measurements should be at least 5° to ensure a real change [12,20,21,22,23]. However, our research shows that exiguous measurement changes (at most 0.5°) are representative. This aspect could be related to the five-hour training sessions before measuring and the experience gained from performing the three measurement rounds with the software prior to the conventional manual measuring.

A comparison of the validity and reliability results of the Cobb angle on AIS X-ray between different studies is difficult due to the diversity of criteria in their design (different number of X-rays, observers, number of measurement sessions, number of weeks between measurement sessions, pre-selection of the limit vertebrae, or measurement tools used) and due to the format of the results (intra- or inter-observer values, ICC, 95%CI for the mean, or only SD). The reliability analyses between the computer-aided and manual measurement produced ICC > 0.99 with 95% CI: 0.996–0.999. The MCD95 was < 0.4°. In our research, we have employed the criteria of absolute reliability [45], which requires a minimum of 30 cases measured by at least six blinded observers with at least three tests per observer, separated from each other by at least two weeks. Different rulers, variable width markers/pencils, and poor-quality X-rays have also been reported as causes of intrinsic error [12,17,32,35], so we controlled for these variables in our study.

In our study, the observers measured the same scoliotic curves with the software and manually to obtain a more meaningful comparison of both measurement methods’ validity and reliability results. In the study, we set a test-retest time of one month so that the variability of the measurements could not be attributed to the observers remembering the measured radiographs nor the results obtained on them. There were no statistically significant improvements between the successive tests because the training sessions avoided common measurement errors from the beginning of the study. The second possible explanation is that when the precision error is so small (less than one degree), it is difficult to improve the precision with training.

The computer-assisted measurement of the Cobb angle eliminates the sources of intrinsic error [12,30,31,32,34] except the selection of the terminal vertebrae reference points of the scoliotic curve, which is supposed to improve the accuracy of manual measurements since the software allows zooming in on the points of interest and varying the brightness of the medical image for better visualisation and, therefore, better selection of the points [42].

Other authors have reported similar error values to ours in determining the Cobb angle using computer systems. These studies were also designed to avoid intrinsic causes of measurement error by considering a large sample of subjects, several observers, and several measurements repetitions. For instance, Srinivasalu et al. [31] and Zhang et al. [38] considered 318 and 60 X-rays, respectively, measured by three observers, three and two times, respectively, obtaining similar MBE and SD values to those of our study. In contrast to our study, these authors [31,38] did not compare their results with manual measurements obtained under the same conditions.

The value of our study lies in the fact that the developed measurement software reproduces the manual measurement method with minimal computer intervention but eliminates some sources of intrinsic error; following the same methodology and using the same subjects and observers, we have studied the error of the manual Cobb angle measurement method. This methodology allows a better comparison between the methods’ (software and manual) validity. In addition, we considered the Hopkins criteria [45] for calculating reliability as well as two experience groups.

We consider that the methodology followed in the manual measurements (reproduction of the performance of the software in the selection of the most tilted vertebrae) limits the error in the selection of the terminal vertebrae.

From the standpoint of statistical inference, it was necessary to treat the values obtained by the observers and the error distributions to reduce the error in any statistical estimation. If we consider outliers, the DCM results were as follows: intra-observer error for the expert group for the software measures was MCD95 = 0.54° and MCD95 = 0.36° for the manual measures; intra-observer error for the novice group for the software measures was MCD95 = 0.45° and MCD95 = 0.64° for the manual measurements. When measuring with the software, the inter-observer error was MCD95 = 0.42° and MCD95 = 0.49° when measuring manually. These values lead us to believe that eliminating outliers does not produce a significant bias.

To avoid bias in the measurements, we established the procedure to follow by employing training sessions for the observers, distinguishing their level of experience, using a sample of subjects sufficiently representative of the population, and considering the temporal stability of the measurements by repeating them at different times.

There are some limitations to our study. First, to establish the “gold standard” and compare manual and software measurements, we used the mean value of each measurement distribution, which means that each measurement may contain a small error. Second, we did not consider each evaluator’s computer equipment (e.g., viewable image size, display resolution, luminance, and contrast ratio or the characteristics of the mouse or touchpad), which may have influenced the accuracy of the measurements. However, the obtained results (error fewer than 2 degrees) seem to be of little significance and would not preclude extrapolating the results to another population of observers with different computer equipment. Third, we did not consider the effect on measurements obtained manually if they had not been measured beforehand with the TraumaMeter software (we think that the results of manual measurements were improved by the previous learning of measurement with the software). Fourth, although we designed our study to meet the Hopkins criteria, which require a minimum of 30 subjects, we suppose that a larger sample of AIS radiographs would have decreased the likelihood of type II error in our results. Fifth, there was a restriction in selecting the radiographs in terms of the severity of the scoliotic curves, as those with a magnitude of less than 10° were discarded. It was also difficult to obtain radiographs with very severe curves, so the number studied in this severity group was small. Finally, the outliers eliminated in each distribution used in the study may be due to imperfect measurement and errors in recording the value of the measurements in the database provided by each observer. However, they accounted for only 2.44% of the total measurements made (41 of 1680 measurements).

These limitations notwithstanding, the authors believe that the study’s outcomes are valuable. Due to its high validity and reliability, the TraumaMeter v.873 software can be recommended for quantifying AIS curves in clinical practice and research.

## 5. Conclusions

Intra-observer measurement errors are lower when using the software TraumaMeter (MBE = 1.8°, SD = 0.65°) than when using the conventional manual Cobb angle measurement method (MBE = 2.31°, SD = 0.83°). The MBE value of the inter-group (expert and novice) distributions is statistically different when using TraumaMeter or the manual method. The error in the measurements depends on the observer skill levels or experiences. The use of the software reduces the difference in error between the novice and expert observers in a statistically significant way. The minimum detectable change (MDC95) is equal to or less than 0.5°, irrespective of the observer’s experience and measurement method (TraumaMeter or manual). There is almost a perfect agreement between the TraumaMeter measurement and the manual method.

## Figures and Tables

**Figure 1 ijerph-19-04655-f001:**
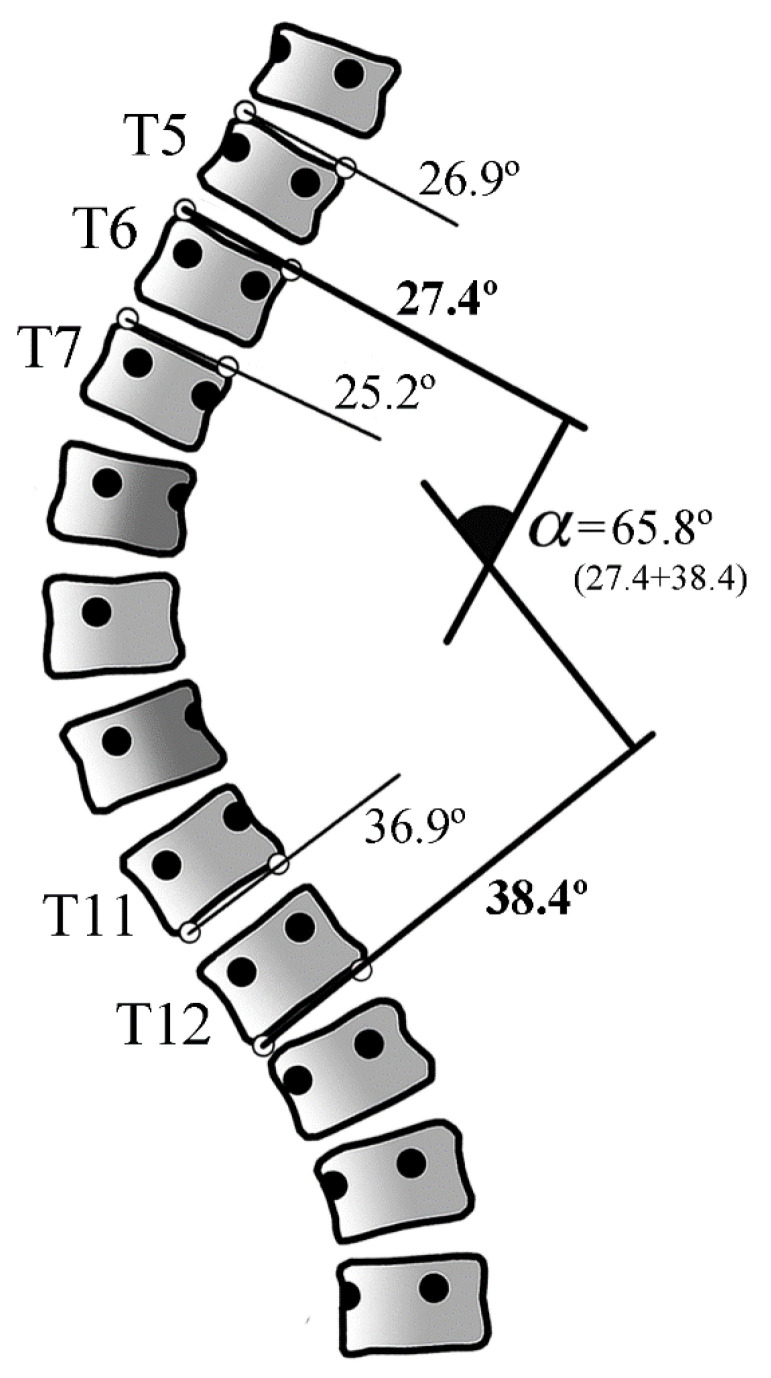
Several vertebrae points can be selected when there is doubt about which vertebrae are more tilted. The software will automatically choose the vertebrae that are most inclined to the horizontal (in this example, T6 (27.4°) and T12 (38.4°)). α: Cobb angle.

**Figure 2 ijerph-19-04655-f002:**
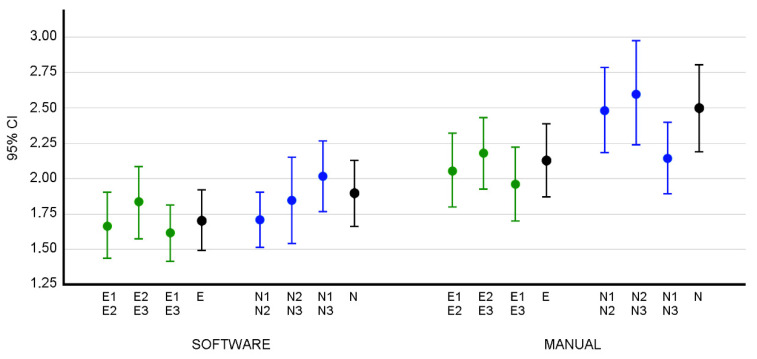
The 95% confidence intervals of the intra-group MBEs. The letter E identifies the measurements obtained by the group of expert observers. E1, E2, and E3 represent the measurements obtained by the group of expert observers in the first, second, and third rounds of measurements, respectively. The letter N identifies the measurements obtained by the group of novice observers. N1, N2, and N3 represent the measurements obtained by the group of novice observers in the first, second, and third rounds of measurements, respectively. The intervals for MBE in the error distribution of E1E2 (between the first and second round of expert measurements), E2E3, E1E3, and E (interval for the intra-group MBE when considering the three batches of expert measurements) are shown. In the same way, the intervals for the different measurement runs of the novice group are shown. Both distributions are shown for the data obtained both with the software and manually, where E and N are the intra-group error distributions in the three measurement runs of the expert (E) and novice (N) groups. In green, the errors of the intra-group measurements of the Expert group between measurement rounds 1 and 2, 2 and 3 and 1 and 3. In blue, the errors of the intra-group measurements of the Novice group between measurement rounds 1 and 2, 2 and 3 and 1 and 3. In black, the errors in the measurements of the Expert and Novice groups in all three tests.

**Figure 3 ijerph-19-04655-f003:**
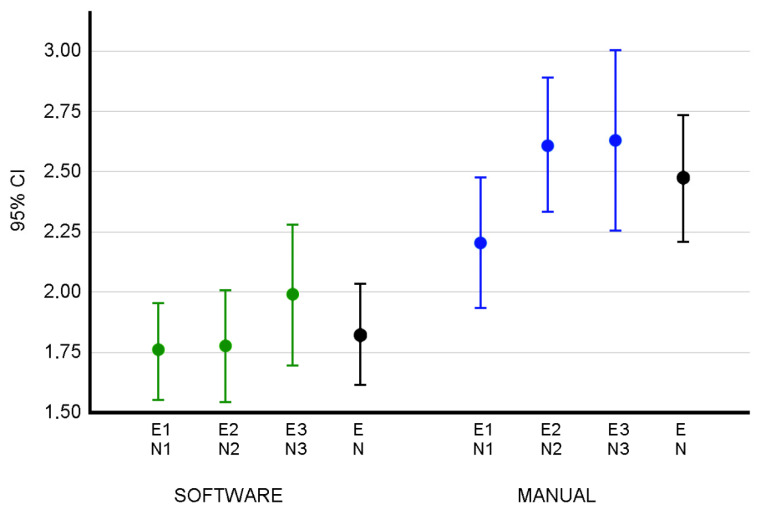
The 95% confidence intervals of the inter-group MBEs. The letter E identifies the measurements obtained by the group of expert observers. E1, E2, and E3 represent the measurements obtained by the group of expert observers in the first, second, and third rounds of measurements, respectively. The letter N identifies the measurements obtained by the group of novice observers. N1, N2, and N3 represent the measurements obtained by the group of novice observers in the first, second, and third rounds of measurements, respectively. Intervals are shown for MBE in the error distribution E1N1 (between the first batch of experts and the first round of novices), E2 N2, E3N3, and EN (interval for the inter-group MBE when considering the three rounds of expert and novice measurements). Confidence intervals are shown for the error distributions of the measurements obtained both with the software and manually, where EN is the distribution of inter-group errors in the three measurement rounds of the expert (E) and novice (N) groups. In green, the inter-group measurement errors when measuring with the software between measurement rounds 1, 2 and 3. In blue, inter-group errors when measuring manually between measurement rounds 1, 2 and 3. In black, the inter-group errors when considering the set of the three tests.

**Figure 4 ijerph-19-04655-f004:**
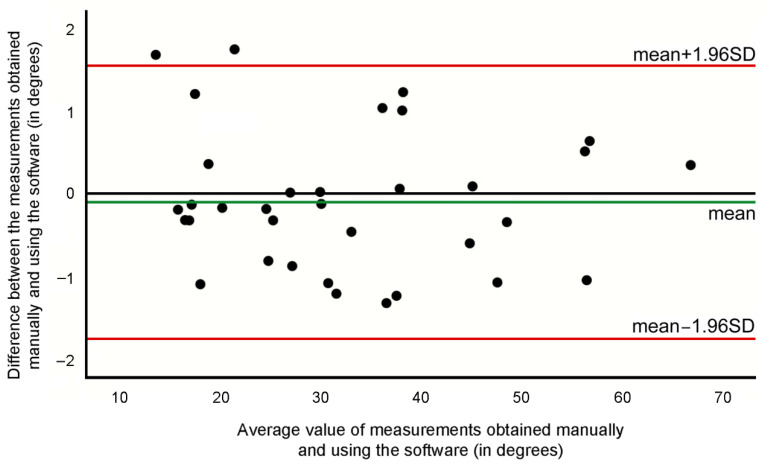
Bland–Altman graphic for the curves’ measurements acquired with the software and manually.

**Table 1 ijerph-19-04655-t001:** The intra- and inter-group validity and reliability analysis with the software and manual measures.

Intragroup Analysis with Software								Intergroup Analysis with Software							
	MBE	SD	gl	SE	MDC95	ICC (2,1)	CI 95%		MBE	SD	gl	SE	MDC95	ICC (2,1)	CI 95%
**E1E2**	1.67	0.67	34	0.11	0.32	0.987	0.978–0.993	**E1N1**	1.75	0.57	33	0.10	0.27	0.983	0.972–0.991
**E2E3**	1.83	0.74	35	0.13	0.35	0.984	0.974–0.991	**E2N2**	1.77	0.65	33	0.11	0.32	0.975	0.959–0.987
**E1E3**	1.61	0.56	33	0.10	0.27	0.986	0.976–0.992	**E3N3**	1.99	0.84	34	0.14	0.40	0.981	0.969–0.99
**E**	1.71	0.61	34	0.11	0.29	0.986	0.977–0.992	**EN**	1.82	0.59	33	0.10	0.29	0.973	0.954–0.987
**N1N2**	1.71	0.55	32	0.10	0.27	0.971	0.952–0.985								
**N2N3**	1.85	0.87	34	0.15	0.41	0.970	0.950–0.984								
**N1N3**	2.02	0.71	34	0.12	0.34	0.977	0.962–0.988								
**N**	1.90	0.67	34	0.12	0.32	0.970	0.950–0.985								
**Intragroup Analysis with the Manual Method**								**Intergroup Analysis with the Manual Method**							
	**MBE**	**SD**	**gl**	**SE**	**MDC95**	**ICC (2,1)**	**CI 95%**		**MBE**	**SD**	**gl**	**SE**	**MDC95**	**ICC (2,1)**	**CI 95%**
**E1E2**	2.08	0.74	35	0.13	0.35	0.982	0.971–0.990	**E1N1**	2.20	0.77	34	0.13	0.37	0.975	0.959–0.987
**E2E3**	2.08	0.73	34	0.12	0.35	0.978	0.964–0.987	**E2N2**	2.61	0.81	35	0.14	0.38	0.974	0.955–0.987
**E1E3**	1.96	0.75	34	0.13	0.36	0.982	0.972–0.990	**E3N3**	2.63	1.05	33	0.18	0.50	0.976	0.961–0.987
**E**	2.13	0.75	35	0.13	0.35	0.981	0.970–0.990	**EN**	2.47	0.76	34	0.13	0.36	0.973	0.951–0.988
**N1N2**	2.49	0.84	33	0.15	0.41	0.967	0.944–0.984								
**N2N3**	2.61	1.07	35	0.18	0.50	0.976	0.958–0.988								
**N1N3**	2.15	0.69	31	0.12	0.34	0.974	0.955–0.987								
**N**	2.50	0.88	34	0.15	0.42	0.974	0.954–0.988								

AXBY is the distribution of errors between the measurements of experience groups A and B in tests X and Y. E stands for experts and N for novices. MBE is the mean bias error, SD is the standard deviation, gl is the number of sample measurements (gl = 35 − outliers), SE is the standard error of the sample, MDC95 is the minimum detectable change (in degrees), ICC (2,1) is the intra-class correlation coefficient of absolute concordance, and CI 95% is the 95% confidence interval.

## Data Availability

Not applicable.

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
