# Peer review of "Validity and Absolute Reliability of the Cobb Angle in Idiopathic Scoliosis with TraumaMeter Software"

_ijerph, 2022, doi:10.3390/ijerph19084655_

Round 1

Reviewer 1 Report

I don't have critical comments. I recommend publishing in IJERPH. 

Author Response

We are grateful for your recommendation that our manuscript is published.

Reviewer 2 Report

All in all, this paper presents a robust evaluation study of a software-based cob angle measurement. The design of the evaluation is good, but comparable software solutions have already been published several times with comparably good results to manual measurements, partly methodically more convincing. In the end, this well-designed study is a useful addition to the scientific evaluations of software-based measurements under special task conditions and can further contribute to lowering the inhibition threshold for the use of these software solutions. Therefore, I support a publication of this study. 
From my personal point of view, there are still a few points standing in the way of a publication, which are presented in the following:

  • line 84: The imprecise wording less than 2° or 3° is scientifically difficult. Please decide on a value and justify it with valid literature.

  • 132-134: This study investigates the ability of software to measure the cob angle in an X-ray image. However, you define as a group of experienced users only one physician who is more regularly confronted with this measurement method and three physiotherapists. No spine surgeon appears to be involved. From personal experience, I have some reservations about this selection of experts - physiotherapists often have different professional preconceptions when processing x-rays. In comparable studies, this selection is defined as "inexperienced users". They provide a large target here for the audience to whom they are presenting this study - which is primarily physicians.... If this grouping is kept like this it must be addressed as a limitation in the discussion.  

  • 317-320: An important point is made here where one aspect is missing. Besides the fact that there is a training effect when the software is used first and the manual measurement is performed later, this effect is also intensified when the same images are used. It should be explained why all subjects processed the same images - the effect could have been avoided or at least reduced if more images were used and the subjects measured different images using software and manually. 

  • 261-266 and 323-324: The MCD value in this study is very low. This is to some extent due to the fact that the outliers were removed. The reason for this is well presented and sufficiently explained. However, the impact on the calculated MCD value needs to be thoroughly presented to overcome the impression of a relevant bias. From my point of view, it is not sufficient to state the relatively small number of excluded measurements, they should also calculate an MCD value considering all measurements. If this also turns out to be below the previously defined limit, it would present your data in a better light. Please provide this data in the context of the discussion.

Basically, 2 aspects should be addressed really extensively in the discussion in order not to make the publication questionable:
The low number of cases - the impression is created that the minimum acceptable number of X-ray images was used and the unusual selection of "experts" for cob angle measurements. 

Author Response

All in all, this paper presents a robust evaluation study of a software-based cob angle measurement. The design of the evaluation is good, but comparable software solutions have already been published several times with comparably good results to manual measurements, partly methodically more convincing. In the end, this well-designed study is a useful addition to the scientific evaluations of software-based measurements under special task conditions and can further contribute to lowering the inhibition threshold for the use of these software solutions. Therefore, I support a publication of this study.

First of all, we would like to thank you for your comments and allow us to address the questions you raised to improve the manuscript’s quality. We greatly appreciate your observations and the time you have spent on constructive critique and feedback on our manuscript.

From my personal point of view, there are still a few points standing in the way of a publication, which are presented in the following:

We will reply to your point by point, indicating the modifications we propose to the manuscript to adapt it to your comments.

line 84: The imprecise wording less than 2° or 3° is scientifically difficult. Please decide on a value and justify it with valid literature.

The papers consulted on computer-assisted measurement systems to measure Cobb’s angle on AIS images using different strategies show intra-observer errors varying between 1° and 4° (Shea, 1998; Kuklo, 2006; Wills, 2007; Srinivasalu, 2008; Zhang, 2010; Ricart, 2011 and Chan, 2014).

With this in mind, we propose to rewrite the paragraph as follows (lines 87 to 91): “We have hypothesised that the intra-observer error in Cobb’s angle measurements is less than 2.5° in Novels and even better (less than 2°) in Experts and that the use of the software TraumaMeter improves the validity and reliability of the measurements obtained compared to the manual method.”

132-134: This study investigates the ability of software to measure the cob angle in an X-ray image. However, you define as a group of experienced users only one physician who is more regularly confronted with this measurement method and three physiotherapists. No spine surgeon appears to be involved. From personal experience, I have some reservations about this selection of experts - physiotherapists often have different professional preconceptions when processing x-rays. In comparable studies, this selection is defined as "inexperienced users". They provide a large target here for the audience to whom they are presenting this study - which is primarily physicians.... If this grouping is kept like this it must be addressed as a limitation in the discussion. 

We find your observation very interesting and incorporate it in the Discussion section as a study limitation (lines 351 to 357):  

“Fourthly, it is a debatable decision that the division of the observers into Expert and Novice groups was made based on the frequency with which they use the Cobb method rather than based on their medical speciality. We must point out that those we have considered Experts are not spine surgeons. Still, one of them is an orthopaedic surgeon. The other two are doctors specialising in Rehabilitation and Physical Medicine, who in our environment are very often involved in the follow-up and monitoring of patients with AIS.”

In our study, we considered Experts, those observers accustomed to measuring spinal misalignments in their daily practice. These observers are part of the same research group at the University of Murcia, with more than 30 years of experience in spinal misalignments, and there is homogeneous training among them. As indicated in the new paragraph to be incorporated as a limitation, those observers that we have considered Experts are not spine surgeons. Still, one of them is an orthopaedic surgeon. The other two are MD specialising in Rehabilitation and Physical Medicine, who in our environment are very often involved in the follow-up and monitoring of patients with AIS.

317-320: An important point is made here where one aspect is missing. Besides the fact that there is a training effect when the software is used first and the manual measurement is performed later, this effect is also intensified when the same images are used. It should be explained why all subjects processed the same images - the effect could have been avoided or at least reduced if more images were used and the subjects measured different images using software and manually.

The number of radiographs was determined using the Hopkins criteria (minimum of 30 cases, six blinded observers and at least three tests per observer, separated by at least two weeks) (Hopkins, 2000 and Atkinson, 2001) and noting that the measured curves had different degrees of AIS (low, moderate, severe, and very severe scoliosis).

Taking this into account, radiographs with equivalent image quality and no defects were selected, representing 35 scoliotic curves (subjects) that fulfilled the requirements mentioned earlier.

Indeed, one of the sources of random error in the measurements of each subject (each curve of a radiograph) is the temporal stability of the measurement method, which depends, among other factors, on the observers. In our study, we considered that the hasty application of the retest may affect the internal validity due to the possible recall of previous measurements. Thus, we established a test-retest time of one month. In this time interval, the variability of the measurements cannot be attributed to the observers remembering the measured radiographs, nor the results obtained in them.

The authors considered that due to the characteristics of our study, the one-month period is also not so long as to affect the internal validity through the temporal stability of the measurements.

In addition, to avoid bias when measuring the same radiographs, the radiographs were sent to each measurer in random order and without any markings to identify them. If we add that the measurer did not have access to the previous radiographs, it is understood that all measurements were performed in a blinded manner.

The same images were used in both studies, with the TraumaMeter software and with manual measurements. This way, a more significant comparison of validity and reliability results between the two measurement tools is possible. We believe this is an improvement over other published studies, where this comparison is made with data obtained using both measurement tools (computer-assisted measurement software and manual) using different measurement methods (different subjects, observers, time between tests, methodology), etcetera.

We have added to the manuscript this aspect that you mention in the discussion section in lines 294 to 298:

“In our study, the observers measured the same scoliotic curves with the software and manually to obtain a more meaningful comparison of both measurement methods’ validity and reliability results. In the study, we set a test-retest time of one month so that the variability of the measurements could not be attributed to the observers remembering the measured radiographs nor the results obtained on them.”

261-266 and 323-324: The MCD value in this study is very low. This is to some extent due to the fact that the outliers were removed. The reason for this is well presented and sufficiently explained. However, the impact on the calculated MCD value needs to be thoroughly presented to overcome the impression of a relevant bias. From my point of view, it is not sufficient to state the relatively small number of excluded measurements, they should also calculate an MCD value considering all measurements. If this also turns out to be below the previously defined limit, it would present your data in a better light. Please provide this data in the context of the discussion.

If outliers are considered, the DCM results obtained are as follows:

* When studying the intra-observer error for the group of Experts when measuring with the software: MCD95=0.54°.

* When studying the intra-observer error for the group of novices when measuring with the software: MCD95=0.45°.

* When studying the intra-observer error for the Expert group when measuring manually: MCD95=0.36°.

* When studying the intra-observer error for the group of novices when measuring manually: MCD95=0.64°.

* When studying the inter-observer error when measuring with the software: MCD95=0.42°.

* When studying the inter-observer error when measuring manually: MCD95=0.49°.

We add the requested information to the discussion section (lines 329 to 335):

“If we consider outliers, the DCM results are as follows: Intra-observer error for the Expert group for the software measures MCD95 = 0.54° and MCD95 = 0.36° for the manual measures; Intra-observer error for the Novice group the software measures MCD95 = 0.45° and MCD95 = 0.64° for the manual measurements. When measuring with the software, the inter-observer error was MCD95 = 0.42° and MCD95 = 0.49° when measuring manually. These values lead us to believe that eliminating outliers does not produce a significant bias.”

These results are because the outliers differ significantly from the real values in the few outliers we have obtained. For example, when measuring curve 21b, Expert measurer number 2 entered a value of 7.5° in the excel file, while the average value of all the measurements (considering the outliers) for this curve is 32.1°. Such different data from the real deal was probably because the measurer entered the measure in an erroneous cell of the excel table (corresponding to another curve).

Although we designed our study to meet the Hopkins criteria, which require a minimum of 30 subjects, we suppose that a larger sample of AIS radiographs would have decreased the likelihood of type II error in our results. We incorporate this observation, also as a limitation of the study (lines 358 to 360).

As noted above, we qualify our selection of experts as a potential weakness of the study (lines 351 to 357).

Thank you very much for your comments, which improve the quality of the manuscript!

Reviewer 3 Report

Congratulations to the authors for this manuscript with the aim of assessing the validity and reliability of the TraumaMeter software for measuring the Cobb angle. Please find several comments and suggestions below in order to try to improve the quality of the work.

Introduction: it is well written and provides the most important information. However, please consider to add some information about the problem of study: why is important to test the validity and reliability of these software? Have they been tested before? This issue could be added to the introduction section, prior to the aim of the study.

Methods:

- The X-ray images analyzed were randomly selected?

- There were any restriction for selecting the images regarding the severity of the scoliosis?

Results:

Figure 1 and 2, please clarify the meaning of E, E1, E2 in the figures or in the caption.

Discussion:

- Why authors think that no significant improvements were found with practice (neither in manual nor in software measurements)? This point should be better stated in the discussion section.

Author Response

Congratulations to the authors for this manuscript with the aim of assessing the validity and reliability of the TraumaMeter software for measuring the Cobb angle. Please find several comments and suggestions below in order to try to improve the quality of the work.

We are very grateful that you consider our research to be of interest. We greatly appreciate your observations and the time you have spent on constructive critique to improve the quality of the work.

Introduction: it is well written and provides the most important information. However, please consider to add some information about the problem of study: why is important to test the validity and reliability of these software? Have they been tested before? This issue could be added to the introduction section, prior to the aim of the study.

The Cobb angle is the most important measurement in AIS radiographs, as it is necessary to establish the diagnosis and, consequently, decide on the treatment. A sufficiently large error in the measurement of the Cobb angle can mean, for example, that the indicated treatment varies from orthopaedic (bracing) to surgical (instrumented arthrodesis).

The TraumaMeter software is a measurement tool to measure variables of clinical interest in medical imaging. Our manuscript studies the part of the software used to measure the Cobb angle in AIS radiographs and whether the software measures what it claims to measure and can be safely used to obtain the Cobb angle in AIS radiographs.

In addition to the accuracy and reliability values, we have validated this software by comparing its results with the standard measurement method, which is the manual method.

As a result, we have concluded that this software can be safely used in the case of the Cobb angle in scoliosis.

Following your recommendation, we have included the following paragraph in the introduction section (lines 57 to 61): “The Cobb angle is the most important measurement next to vertebral rotation on AIS radiographs [3,4], as it is necessary to establish the diagnosis and decide on the treatment. A sufficiently large error in the Cobb angle measurement can mean, for example, that the indicated treatment varies from observation to orthopaedic therapy or from bracing to instrumented arthrodesis.”

Methods:

- The X-ray images analyzed were randomly selected?

The radiographs used in the study were randomly selected from different groups of radiographs with curves of similar magnitude. All had equivalent image quality without defects (to avoid a possible source of intrinsic error in the measurements). The initial group of radiographs was studied jointly by two medical specialists with extensive experience in AIS.

- There were any restriction for selecting the images regarding the severity of the scoliosis?

There was a restriction in selecting the radiographs in terms of the severity of the scoliotic curves, as those with a magnitude of less than 10° were discarded. It was also difficult to obtain radiographs with very severe curves, so the number studied in this severity group was small.

We have added to the text this aspect that you mention as a limitation of the study in lines 360 to 364.

Results:

Figure 1 and 2, please clarify the meaning of E, E1, E2 in the figures or in the caption.

The letter E identifies the measurements obtained by the group of Expert observers.

E1, E2 and E3 represent the measurements obtained by the group of Expert observers in the first, second and third rounds of measures, respectively. We have added this clarification to the figure legend.

Discussion:

- Why authors think that no significant improvements were found with practice (neither in manual nor in software measurements)? This point should be better stated in the discussion section.

Before starting the measurements, both (software and manually), a 5-hour session was held in which the study participants were taught how to measure correctly with each of the tools. Perhaps these training sessions made it possible to avoid specific recurring errors in the Cobb angle measurements from the beginning of the study. The second possible explanation is that when the precision error is so small (less than one degree), it is difficult to improve the precision with training.

On the other hand, in the study’s design, we established a test-retest time of one month. After this time interval, there would be no variability in the measurements attributed to the observers remembering the radiographs or the results obtained in them.

We have added to the manuscript this aspect that you mention in the discussion section in lines 298 to 302:

“There are no statistically significant improvements between successive tests because the training sessions avoided common measurement errors from the beginning of the study. The second possible explanation is that when the precision error is so small (less than one degree), it is difficult to improve the precision with training.”

Thank you very much for your comments, which improve the quality of the manuscript!

Round 2

Reviewer 2 Report

Dear authors, I really appreciated the detailed revision. The study can be published very well in this form. 

I have the impression that two points of mine were perhaps formulated a bit too aggressively and would therefore like to add the following:

1. in my view, the composition of the expert group does not necessarily have to consist of spine surgeons. Only an explanation of why the selected persons were defined as experts was necessary to make this selection not intangible. The present addition that these persons are regularly involved in the follow-up treatment of scoliosis patients is sufficient; I would recommend deleting the explicit reference to missing spine surgeons before final publication. This criticism of your own work is unnecessary. 

2. The question as to why more radiographs were not used. The Hopkins criteria were correctly fulfilled with the number of x-rays used - however, the Hopkins criteria define the minimum requirements, from my point of view it would have been beneficial for the validity to go beyond these minimum requirements... Nevertheless, I find the argumentation you put forward in this regard in its current form plausible and more comprehensible and thus also well acceptable. No further changes are required from my point of view.

Congratulation to the authors on this study and on the publication, which should now be completed.

Author Response

Again, thank you very much for your comments.

We have modified the manuscript, eliminating the selection of Experts and Novices as one of the limitations and have added in the Material and Method Section the following paragraph in order to state the assignment of roles to the observers clearly:

“The division of the observers into Expert and Novice groups was made based on the frequency with which they use the Cobb method rather than based on their medical speciality. We have considered Experts observers very often involved in the follow-up and monitoring of patients with AIS.”